# The Spectral Footprint of Neural Activity: How MUAP Properties and Spike Train Variability Shape sEMG

**DOI:** 10.3390/bioengineering12111181

**Published:** 2025-10-30

**Authors:** Alvaro Costa-Garcia, Akihiko Murai

**Affiliations:** Research Institute on Human and Societal Augmentation, National Institute of Advanced Industrial Science and Technology (AIST), Kashiwa 277-0882, Japan

**Keywords:** electromyography, sEMG spectrum, muscle modeling, muscle activity

## Abstract

Surface electromyographic (sEMG) signals result from the interaction between motor unit action potentials (MUAPs) and neural spike trains, yet how specific features of spike timing shape the sEMG spectrum is not fully understood. Using a simplified convolutional model, we simulated sEMG by combining synthetic spike trains with MUAP templates, varying firing rate, temporal jitter, and motor unit synchronization to examine their effects on spectral characteristics. Rather than addressing a particular experimental condition such as fatigue or workload, the main goal of this study is to provide a framework that clarifies how variability in neural timing and muscle properties affects the observed sEMG spectrum. We introduce extractability indices to measure how clearly neural activity appears in the spectrum. Results show that MUAPs act as spectral filters, reducing components outside their bandwidth and limiting the detection of high firing rates. Temporal jitter spreads spectral energy and blunts frequency peaks, while moderate synchronization improves spectral visibility, partially countering jitter effects. These findings offer a reference for interpreting how neural and muscular factors shape sEMG signals, supporting a more informed use of spectral analysis in both experimental and applied neuromuscular studies.

## 1. Introduction

Surface electromyographic (sEMG) signals represent the electrical activity of active muscle fibers during contraction, as recorded on the skin surface. The temporal and spectral features of these signals are shaped by a wide array of both physiological and non-physiological factors, including the discharge characteristics of motor neurons, the shape and variability of motor unit action potentials (MUAPs), the spatial distribution of muscle fibers, volume conduction, electrode configuration, and interference phenomena such as amplitude cancelation due to the superposition and synchronization of motor units. These factors and their influence on sEMG signal interpretation have been extensively discussed in a series of reviews by Farina et al. [1,2,3,4], particularly in the context of evaluating the feasibility of extracting neural control strategies from surface recordings. These works emphasize the inherent limitations of such inference, given the heterogeneity and interdependence of the contributing variables.

The large number of parameters that influence the sEMG signal, combined with the practical impossibility of accurately measuring or controlling all of them simultaneously, make it difficult to approach the problem from a fully integrative standpoint. In such situations, isolating coherent subsets of influencing factors and evaluating their effects under idealized conditions may offer a more tractable approach, providing general insight into their specific roles. The present study adopts this strategy by focusing on a subset of physiological variables.

The analysis centers on the frequency-domain transformation that occurs between the motor neuron spike train and the resulting sEMG signal. In particular, attention is given to how the spectral content of the spike train interacts with the spectral shape of the MUAP waveform, either preserving or masking neural frequency components in the final signal. This interaction is modeled from a signal-processing perspective, in which the sEMG is described as the temporal convolution of a spike train with a MUAP, and the resulting spectral consequences are examined. Although this convolutional relationship has been acknowledged in conceptual models of sEMG generation [5], its practical implications, specifically the conditions under which neural frequencies become detectable in the sEMG, remain poorly understood.

Clarifying how physiological variability modulates this spectral transformation is essential for determining when the sEMG can reliably reflect central neural activity. Although previous studies have investigated how cortical oscillations propagate to spinal motor neurons [6], less is known about how these signals survive the neuromuscular filtering process and become observable at the muscle level. Identifying the neuromuscular conditions that facilitate or hinder the spectral expression of neural drive in sEMG is relevant for interpreting corticomuscular coherence, tracking fatigue-related spectral shifts, and designing robust EMG decomposition algorithms. More fundamentally, such insights inform the theoretical limits of neural observability in peripheral muscle recordings.

To address this problem, a simplified mathematical model is introduced to formalize how the convolution of two time-domain signals, such as the spike train and the MUAP, leads to a blending of their frequency contents. The MUAP is influenced by physiological properties such as fiber conduction velocity, fiber radius, and intracellular conductivity, as well as extrinsic factors like spatial filtering due to volume conduction and electrode placement.

In contrast, the spike train is influenced by different factors, including temporal jitter, firing rate, and synchronization between motor units. Given the focus of this study on how the spectral content of the spike train is preserved or masked in the sEMG spectrum, the analysis systematically varies these parameters to examine their impact on the spike train structure. Meanwhile, a simplified approach is adopted for the MUAP, where only changes in its bandwidth are considered, capturing the general spectral effects associated with variations in conduction velocity and volume conduction’s low-pass filtering properties.

This approach isolates the contribution of neural variability to the spectral transformation, while still accounting for a physiologically relevant range of MUAP spectral properties, without explicitly modeling the complex anatomical and biophysical determinants of MUAP morphology.

## 2. Materials and Methods

### 2.1. Modeling Framework

sEMG signals can be conceptualized as the summation of MUAP trains generated by individual motor units. Each MUAP train can be modeled as the temporal convolution of two components: a spike train representing the motor-neuron firing activity and the characteristic MUAP waveform, which reflects the electrophysiological response of the innervated muscle fibers [5].

To derive the frequency spectrum of an sEMG signal, fundamental Fourier transform properties are applied:Temporal summation and frequency domain summation: Summing multiple signals in time corresponds to summing their spectra in frequency, allowing the analysis of individual motor unit contributions to the overall sEMG.Temporal convolution and spectral multiplication: Convolution in time translates to multiplication in frequency. Here, the spike train convolved with the MUAP waveform produces a MUAP train spectrum by multiplying the spike train and MUAP spectra.

Equation (Equation 1) formalizes this interaction in the frequency domain:(1)$$\begin{array}{cc} \mathrm{sEMG}(t) & =\sum_{MU=1}^{N}ST_{MU}\left(t\right)*MUAP_{MU}\left(t\right)\\ \; & ⇕\mathrm{FFT}\\ \mathrm{sEMG}(f) & =\sum_{MU=1}^{N}ST_{MU}\left(f\right)\cdot MUAP_{MU}\left(f\right) \end{array}$$
where ST(t) and ST(f) are spike trains in time and frequency, MUAP(t) and MUAP(f) are MUAP waveforms in time and frequency, and *N* is the number of contributing motor units.

Figure 1 illustrates how the spike train spectrum, which spans a broad frequency range, is modulated by the MUAP spectrum, concentrated in a specific bandwidth, producing the final sEMG spectrum.

In the following sections, both spike trains and MUAPs are modeled based on parameters affecting their spectral distribution, to test how changes impact the sEMG according to Equation (Equation 1).

No real experimental data were used, but this approach allows precise control over spike timing, jitter, and synchronization, which is challenging or impossible in vivo. By simulating physiologically plausible conditions, this framework provides clear insight into how spike trains and MUAP spectra interact, avoiding the measurement limitations and noise inherent in real sEMG recordings.

### 2.2. Modeling MUAPs

MUAPs represent the summed transmembrane potentials of all muscle fibers innervated by a given motor neuron. In monopolar recordings, the waveform observed at the skin surface typically resembles a modified Ricker wavelet, with asymmetric lateral lobes due to volume conduction and the superposition of multiple fiber action potentials [7]. In the present study, MUAPs are simulated using the model described by Costa et al. [8], applied to a medium-sized biceps muscle (35 cm in length), with the innervation zone positioned at 60% of the distance from the shoulder to the distal tendon [9]. A monopolar electrode configuration is used (as this configuration is standard in high-density sEMG recordings employed for motor unit decomposition and spike-train identification [4]) with the recording site placed centrally between the innervation point and the distal tendon, following standard anatomical guidelines such as those proposed by the SENIAM project [10].

The present implementation explicitly simulates the individual muscle fibers that compose each motor unit and accounts for volume-conductor effects of the tissues and skin. Each fiber action potential is propagated and filtered according to its specific location, depth and orientation before reaching the electrode; thus, the surface potential of a single fiber can be distorted by spatial and conductor effects. Because fibers within a motor unit are activated approximately synchronously, the MUAP recorded on the skin is the temporal superposition of these individually distorted fiber potentials. That superposition tends to smooth out fiber-to-fiber inconsistencies: local distortions partially cancel each other, yielding a regular, smoothed MUAP whose gross shape is principally determined by the ensemble’s mean conduction velocity.

Spatial factors such as fiber depth, inclination, and electrode position continue to influence MUAP morphology via low-pass filtering and amplitude attenuation. To capture both physiological variability and residual spectral effects arising from anatomy and volume conduction, MUAPs are simulated across an extended range of mean conduction velocities (2–6 m/s). This range encompasses typical values for different fiber types (typically 3–5 m/s) while also covering variability introduced by anatomical and recording conditions [11].

Figure 2 illustrates a series of simulated MUAPs and their corresponding spectra across this range. As expected, higher mean conduction velocities produce narrower, sharper MUAPs with broader spectra, whereas lower mean velocities yield more prolonged waveforms with reduced spectral bandwidth.

By explicitly modeling individual fiber positions and the volume conduction environment, including skin and tissue layers, while parameterizing MUAP morphology primarily via the ensemble mean conduction velocity, the model preserves key spatial and conductor effects (including nonlinear distortions at the skin surface) relevant for spectral analysis. Importantly, the effects of volume conduction are resolved using finite element modeling (FEM) [8], which has been proposed as an effective methodology to account for electromagnetic field changes arising from the non-arbitrary distribution of biological tissues [12]. Despite this complexity, the framework remains tractable for the systematic investigation of how spike-train spectra and MUAP bandwidth interact in sEMG signals.

### 2.3. Modeling Motor Neuron Spike Trains

The spike trains of motor neurons are modeled as sequences of Dirac delta impulses. This representation reflects the fact that, in the context of sEMG signal generation, the timing of each neural discharge is the sole determinant of motor unit activation. Although motor neuron action potentials have a finite waveform as they propagate along axons, their only functional role in muscle activation is to trigger the release of acetylcholine (ACh) at the neuromuscular junction. Once the ACh concentration exceeds the activation threshold of the muscle fibers, contraction is initiated. From the perspective of EMG modeling, therefore, the shape of the axonal action potential is irrelevant, and a spike train can be effectively represented as a sequence of idealized impulses, each corresponding to a firing event.

To model the temporal structure of motor neuron activity, each spike train is defined by two key parameters: its average firing frequency and its temporal jitter. This minimal parametrization is chosen deliberately, as the study does not aim to replicate full motor unit behavior (including recruitment thresholds, rate coding, or common synaptic inputs) but instead focuses on the basic spectral consequences of timing variability in the neural drive. The intent is to isolate the influence of temporal dispersion on the detectability of neural frequency content in the resulting sEMG.

Average firing frequency: This parameter defines the average number of impulses generated per unit time by a single motor neuron.Jitter Index: The Jitter Index reflects the degree of temporal variability in spike timing relative to an ideal periodic train. Although neuron inter-spike intervals (ISIs) are often modeled using Poisson statistics in theoretical neuroscience [13,14], physiological recordings on motor neurons suggest that their variability often approximates a Gaussian distribution centered around a preferred mean interval, especially under stable neuromuscular drive [15]. The spike train generation method used in this study linearly interpolates between fully periodic timing and stochastic timing, yielding ISI distributions that transition smoothly from delta-like (JI = 0) to exponential (JI = 1), with intermediate values effectively producing Gaussian-like ISI distributions with increasing variance.

Figure 3 illustrates the effects of the Jitter Index (JI) on motor neuron spike trains, their spectral properties, and the spectral characteristics of the resulting MUAP trains. The 30 Hz example shown here is selected solely for visualization purposes, as it represents the midrange of those frequencies commonly observed during moderate isometric contractions [16]. As shown, increasing JI introduces greater stochasticity into the spike trains, leading to a progressive broadening of the spectral peaks associated with the average firing frequency. For high values of JI, the spectral peaks become indistinct, reflecting the significant variability in spike timing. Additionally, when analyzing the spectrum of the MUAP train, the modulatory effect of the MUAP waveform further distorts the spectral peaks originating from the motor neuron spike train. This highlights the combined influence of spike variability and MUAP shaping on the spectral features of the simulated signal.

### 2.4. Modeling Synchronization Between Motor Units

A function that synchronizes several motor units from their average firing frequency, Jitter Index, and a Synchronization Index (SI) were implemented. This function allows us to explore two distinct configurations of synchronization based on the physiological phenomena observed in motor unit behavior [17].

#### 2.4.1. Firing Frequency-Based Synchronization

In this configuration, synchronization is defined as the similarity in the average firing frequencies of different motor units. The firing frequencies are modeled using a Gaussian distribution centered on the overall average firing frequency of the group, with its standard deviation modulated by the SI. When SI = 0, the standard deviation is maximized (fixed at 5 Hz in this study), resulting in highly dispersed firing frequencies. Conversely, when SI = 1, the standard deviation is reduced to zero, meaning all motor units fire at the same average frequency. This definition captures the degree of uniformity in firing rates without considering the temporal alignment of spikes.

#### 2.4.2. Firing Frequency- and Time-Based Synchronization

In the second configuration, synchronization includes not only uniformity in firing frequencies but also alignment in the spike timing of motor units. To achieve this, individual spike trains are first generated for each motor unit based on their respective average frequencies (determined by the Gaussian distribution) and Jitter Index. Then, spike timing alignment is introduced by adjusting the spike times of all motor units to match a “base” spike train chosen from one of the motor units. The degree of alignment is proportional to the SI:For SI = 0, no alignment occurs, and the spike times remain as initially generated.For SI = 1, all spike trains fully match the base spike train.For intermediate SI values, spikes are partially adjusted toward the base train’s timing.

#### 2.4.3. Motivation for Dual Definitions of Synchronization

These two configurations of synchronization are motivated by experimental observations during sustained isometric contractions. Research indicates that as force stabilizes during such contractions, motor units tend to converge toward similar average firing frequencies. Some studies suggest that this convergence is primarily in firing rates [18], while others argue that common input leads to temporal synchronization in spike times, which also affects firing rates [19]. Although this discussion is not the focus of this work, the potential presence or absence of temporal synchronization in spike trains could significantly influence the identification of key firing frequencies. Therefore, both types of synchronization are implemented. By systematically varying the SI in these models, it will be possible to simulate and analyze the impacts of both frequency and time-based synchronization on the spectral properties of the resulting sEMG signal.

### 2.5. Indices for Assessing Information Extractability from sEMG Signals

The model described above illustrates how the spectrum of the sEMG signal arises from the interaction between the spectral properties of the spike train and the MUAP waveform. Since the MUAP spectrum is primarily shaped by intrinsic electrophysiological characteristics of muscle fibers and volume conduction properties, any information related to the timing or regularity of motor neuron discharges must be reflected in the spike train spectrum. To quantify how prominently each of these components is represented in the resulting sEMG signal, two spectral extractability indices are proposed:Spike train Extractability Index (SEI): Defined as the correlation coefficient between the sEMG spectrum and the spike train spectrum. This index quantifies how strongly the spectral structure of the neural drive is retained in the final signal (see Figure 4A).MUAP Extractability Index (MEI): Analogously, this index computes the correlation between the sEMG spectrum and the MUAP spectrum, representing the extent to which the muscular component dominates the resulting signal (see Figure 4B).

Although a high SEI may indicate that the spike train spectrum is present within the sEMG, this does not necessarily imply that its fundamental frequency components, such as those related to the average firing rate, can be easily identified. Parameters such as temporal jitter (see Figure 3) and spike timing variability can obscure the harmonics of the neural signal. To capture this aspect more explicitly, a third index is introduced:Firing rate Extractability Index (FEI): This index measures the average correlation between the first three harmonic peaks of the spike train spectrum and the sEMG spectrum. It provides a targeted assessment of how prominently the fundamental firing rhythm is preserved in the observed signal (see Figure 4C).

All three indices are bounded between −1 and 1 due to the properties of the correlation coefficient. In this study, negative values are interpreted as an absence of correlation and are set to zero, yielding a range from 0 (no extractability) to 1 (perfect spectral correspondence). It should be noted that the computation of these indices assumes prior knowledge of the spike train spectrum, MUAP waveform, or firing frequency. While such information is typically unavailable when analyzing real-world data, these indices serve as reference tools for comparative analysis under controlled simulation conditions, allowing for the systematic evaluation of spectral transparency.

In addition to these extractability indices, an auxiliary metric is introduced to quantify the degree of spectral randomness in the sEMG signal. This measure reflects the extent to which the spectrum resembles a smooth, broadband distribution, as might arise from high spike timing variability. Increased randomness can degrade the detectability of structured spectral features, such as harmonics or sharp peaks.

Spectral Randomness Index (SRI): Defined as the correlation coefficient between the normalized histogram of the sEMG spectrum and a fitted Gaussian distribution of matching mean and variance (see Figure 4D).

Unlike the previous indices, the SRI does not require any prior knowledge about the underlying neural or muscular sources, making it applicable to experimental data. Although its relationship to extractability indices remains to be fully characterized, preliminary observations suggest that increased spectral randomness is often associated with a reduced prominence of neural features. As such, the SRI may offer a practical, indirect proxy for assessing signal structure and information content in both simulated and real sEMG recordings.

### 2.6. Simulation Protocols

To evaluate the indices defined in this study, two sets of simulations were conducted.

#### 2.6.1. Single Motor Unit Simulations

This first set of simulations was designed to examine the effect of jitter, firing frequency, and the spectral shape of motor unit action potentials (MUAPs) on the extractability indices (SEI, MEI, FEI) and the randomness index (SRI). The following parameter ranges were tested:Jitter Index (JI): Varied from 0 to 1 in increments of 0.01.Firing rate (FR): Firing frequencies from 5 Hz to 60 Hz in increments of 1 Hz.MUAP Spectral Shapes: Represented for conduction velocities from 2 to 6 m/s in increments of 0.5 m/s.

Each simulation was repeated 20 times to account for the inherent randomness introduced by jitter. The indices were calculated as the average across these repetitions to mitigate the possibility of a high JI still resulting in a periodic pattern due to random fluctuations.

#### 2.6.2. Multiple Motor Unit Simulations

To explore the impact of synchronization among motor units, a second set of simulations was performed using 20 motor units, a number that reflects the typical subset of units contributing significantly to sEMG signals in experimental recordings [20,21]. In this framework, each motor unit is treated as a single functional block. This configuration corresponds to a simplified steady-state phase of an isometric contraction, avoiding the additional complexity of recruitment dynamics, rotational activation, or fatigue effects.

Synchronization was examined under two distinct conditions:hlSynchronization affecting the average firing rates: In this condition, synchronization was introduced by aligning the average firing frequencies of different motor units, while keeping their individual spike timing uncorrelated.Synchronization affecting both firing rates and spike timing: In this more constrained condition, synchronization also imposed partial temporal alignment between spikes across different units.

For each of these synchronization conditions, the following parameters were systematically varied across physiologically plausible ranges:Jitter Index (JI): From 0 to 1, in increments of 0.01.Firing rate: From 5 Hz to 60 Hz, in increments of 1 Hz.Synchronization Index (SI): From 0 to 1, in increments of 0.05.MUAP Spectral Range: Simulated via conduction velocities ranging from 2 to 6 m/s, in increments of 0.5 m/s.

As in the first set of simulations, each condition was repeated 20 times to account for the stochastic variability introduced by jitter, and all reported indices correspond to the mean values across repetitions to ensure robust and reliable outcomes.

### 2.7. Modeling Limitations

The present model focuses on the spectral interaction between motor neuron spike trains and MUAPs under controlled conditions. To reduce confounding factors, we assume isometric contractions with electrodes placed according to standard anatomical guidelines (SENIAM), which minimize motion artifacts, cross-talk, and other recording noise. As a result, sources of contamination such as non-uniform electrode-skin impedance, electrode movement, tissue heterogeneity, and intermuscular or bilateral coupling are not explicitly simulated. While these simplifications limit the direct applicability of the model to complex, real-world sEMG recordings, they allow a focused investigation of the effects of spike timing, jitter, and intra-unit synchronization. In experimental settings, we recommend following similar protocols (i.e., careful electrode placement, isometric contractions, and standard filtering) to ensure that these factors remain controlled and that observed spectral behaviors reflect primarily neural and MUAP interactions.

## 3. Results

Figure 5 shows the results from the first set of simulations, which focus exclusively on individual motor units. In this set, the effect of jitter on the extractability indices is evaluated across a range of firing frequencies.

The first row presents four 3D surface plots (Figure 5A–D), corresponding to the Fire Rate Extractability Index (FEI), Spike Train Extractability Index (SEI), MUAP Extractability Index (MEI), and Spectral Randomness Index (SRI). The Jitter Index (JI) is shown on the X-axis, the average firing frequency on the Y-axis, and the index values on the Z-axis. These plots display results averaged across all MUAP types, since prior analyses indicated that MUAP shape has a limited influence on the overall index behavior. For completeness, the corresponding plots for different MUAPs are included in Appendix A (Figure A1).

The first two plots (Figure 5A,B) reveal that metrics associated with spike train characteristics—the FEI and SEI—exhibit low extractability values and are highly sensitive to jitter. While the FEI reaches moderate values at low jitter, it rapidly declines and falls below 0.1 for jitter indices above 0.2. The SEI remains consistently low across all conditions, not exceeding 0.5 even under minimal jitter, suggesting limited overall spectral transfer of the spike train structure to the sEMG. The FEI also shows a reduction at higher firing frequencies, indicating diminished visibility of the fundamental frequency at fast discharge rates.

In contrast, the MEI (Figure 5C) remains above 0.5 across most conditions and increases with jitter, suggesting that higher temporal variability enhances the distinctiveness of MUAP-related spectral components. Slightly higher MEI values are observed at lower firing rates. The SRI (Figure 5D) mirrors the trend of the MEI, rising with jitter and indicating greater spectral dispersion, consistent with reduced neural regularity.

The second row (Figure 5E–G) presents 2D plots showing the relationship between the extractability indices (X-axis) and the SRI (Y-axis). The MEI shows a strong positive correlation with the SRI, while the SEI exhibits a clear negative trend, emphasizing that greater spectral randomness impairs the extraction of spike train-related content but enhances MUAP dominance. The FEI–SRI relationship (Figure 5E) is more complex: although the FEI generally decreases as the SRI increases, the rate of this decline is steeper at lower firing frequencies, indicating greater spectral vulnerability to jitter in low-frequency conditions.

Figure 6 and Figure 7 follow the same structure as Figure 5 but introduce the Synchronization Index (SI) as an additional variable. To accommodate this, average firing frequency is removed as a separate axis by averaging index values across the 5–60 Hz range, based on prior observations that jitter has a more pronounced influence. In these plots, the X-axis represents the JI, the Y-axis the SI, and the Z-axis the index values (SEI, FEI, MEI, SRI). The second row in each figure illustrates the relationship between the extractability indices and the SRI across different synchronization levels.

Figure 6 presents results for synchronization modeled solely as similarity in average firing rates among motor units. The 3D plots (Figure 6A–D) show that this form of synchronization has minimal impact on the SEI, MEI, or SRI, with trends resembling those observed in Figure 5. The MEI remains robust and increases with jitter, while the SEI stays low and decreases further as jitter increases. The FEI (Figure 6A), however, is notably affected by firing rate dispersion, showing a reduction when firing rates differ across units.

The 2D plots (Figure 6E–G) reaffirm the observed correlations between the MEI and SRI (positive) and the SEI and SRI (negative). The FEI (Figure 6E) displays a stronger dependence on the SI, with more aligned firing rates (high SI) favoring greater extractability of the firing rhythm. However, this effect is highly sensitive to jitter, with the FEI rapidly collapsing when JI>0.03.

Figure 7 illustrates results under a stricter synchronization condition, where both average firing rates and spike timing across motor units are partially aligned. The MEI (Figure 7C) and SRI (Figure 7D) exhibit similar behavior to Figure 6, reinforcing the robustness of MUAP-related features and the insensitivity of overall spectral randomness to synchronization. The MEI–SRI relationship (Figure 7G) remains strongly linear and positive.

A noticeable change appears in the SEI and FEI (Figure 7A,B), where spike timing alignment enhances extractability. In the SEI (Figure 7B), this improvement is especially pronounced at high synchronization values, partially offsetting the degradation caused by jitter. This effect is also reflected in the 2D plot (Figure 7F), where curves corresponding to different SI levels show clear separation, indicating that spike train extractability benefits from stronger temporal alignment across units.

## 4. Discussion

### 4.1. Model Overview

The model developed in this study formalizes the spectral interaction that occurs when motor neuron spike trains are convolved with MUAP waveforms (see Figure 1). This operation leads to a multiplicative combination of their respective spectra in the frequency domain, meaning that the sEMG spectrum reflects a modulated version of the spike train spectrum shaped by the spectral envelope of the MUAP. Since MUAPs typically have a limited spectral bandwidth, they impose a band-pass filtering effect that attenuates spike train components outside their range. This helps explain why, in our simulations, MUAP spectral features tended to be more extractable than those of the spike train (Figure 5, Figure 6 and Figure 7C vs. Figure 5A,B, Figure 6A,B and Figure 7A,B).

### 4.2. MUAP and Fire Rate Effect

The modulation imposed by the MUAP spectrum has an important implication: it inherently limits the range of motor neuron firing frequencies that can be effectively represented in the sEMG signal. As shown in Figure 5A,E, extractability generally diminished at higher firing frequencies. This occurs because the harmonics of the average firing frequency are most detectable when they fall within the MUAP’s spectral envelope. Once these harmonics exceeded this range, they were attenuated to the point of becoming difficult to distinguish. Supplementary analyses (see Figure A1 in Appendix A) indicate that, under the conditions tested, the detectable frequency range across different MUAP types typically spanned approximately 5–40 Hz, and in some cases extended to about 60 Hz for MUAPs with broader bandwidths.

### 4.3. Jitter Effect and Randomness

Jitter in spike timing exerted a strong influence on the structure of the resulting sEMG spectrum. As observed in Figure 3, increasing jitter caused a dispersion of spectral energy across frequencies, reducing the sharpness of harmonic peaks and producing a smoother, noise-like spectrum. This effect reduced the detectability of spike train harmonics (lower FEI and SEI), but at the same time enhanced the relative prominence of the MUAP spectral envelope (higher MEI). The Spectral Randomness Index (SRI), introduced in this work, captured this dispersion and correlated well with extractability outcomes—positively with the MEI and negatively with the SEI and FEI (Figure 5E–G, Figure 6E–G and Figure 7E–G). Together, these results show that the SRI effectively captures the transition from structured neural activity to stochastic EMG content, linking jitter-induced desynchronization to measurable losses in spectral extractability.

Importantly, the SRI is a standalone metric that does not require prior knowledge of neural or muscular components, making it suitable for use in both simulation and experimental contexts. While it does not provide a direct measure of spectral source contributions, its strong correlation with extractability indices suggests it may serve as a useful proxy for assessing how much structured neural content remains visible in the sEMG signal under different physiological conditions.

### 4.4. Synchronization Effects

Motor unit synchronization substantially altered the expression of spike-train features in the sEMG spectrum. In single-unit cases, fundamental firing frequencies were often detectable in the absence of jitter (Figure 5A). However, when multiple units were combined, the degree of synchronization—particularly in spike timing—played a critical role in preserving or obscuring the neural spectral features. As shown in Figure 6 and Figure 7, synchronization in firing rates alone had limited impact, whereas synchronization in spike timing noticeably improved spike-train extractability, particularly in the presence of moderate jitter. This was reflected in the separation between curves in SEI and FEI values across different SI levels (Figure 7F).

These findings suggest that the spectral visibility of neural timing information depends not only on the individual firing statistics of motor units, but also on their collective coordination. Under high jitter and low synchronization, the spectral imprint of the spike train became diffuse and difficult to distinguish. Conversely, even modest synchronization in spike timing could partially recover extractability lost to jitter.

Hence, Figure 6 and Figure 7 reveal a fundamental trade-off: while jitter introduces noise-like dispersion, partial synchronization can restore structured spectral content, illustrating how coordinated neural activity can enhance EMG interpretability.

### 4.5. Implications for Spectral Analysis of sEMG

Several comprehensive reviews have emphasized the limitations of inferring motor control strategies from surface EMG signals, due to the complex interplay of physiological and non-physiological factors that shape the signal’s properties [1]. These concerns have been further discussed in the Point:Counterpoint debate on whether spectral features of sEMG can provide reliable information about motor unit recruitment or muscle fiber-type composition. The prevailing view, notably argued by Farina and collaborators, is that the spectral content of the sEMG cannot be unambiguously linked to specific neural control strategies or histological properties of the active motor units [22,23].

The results presented in this study provide quantitative support for this view, at least with respect to the neural component of the sEMG. Spectral features originating from the spike train of motor neurons were found to be highly sensitive to physiological variability, particularly spike timing jitter and inter-unit synchronization. Under most of the tested conditions, the spectral imprint of the spike train was substantially altered, which may limit its detectability in the sEMG. These findings are consistent with the idea that methods based on time-domain decomposition of motor unit activity are generally better suited for investigating neural control [4].

Interestingly, the spectral characteristics of the MUAP waveform appeared comparatively more resilient to such variability, suggesting that the muscular component of the sEMG spectrum may remain detectable under a broader range of conditions. However, any attempt to interpret MUAP-related spectral features must be approached with caution. The shape of the MUAP reflects the influence of multiple physiological factors, including conduction velocity, fiber diameter, and intracellular conductivity, but also depends on non-physiological determinants such as electrode configuration and volume conduction spatial filtering.

Taken together, these findings provide further evidence of the limitations of spectral analysis for recovering detailed neural strategies. At the same time, they point to the potential value of exploring the spectral footprint of the MUAP itself, particularly under controlled conditions where confounding variables can be better understood or constrained.

## 5. Conclusions and Future Work

This study analyzed how motor neuron spike trains and MUAP waveforms interact to shape the spectral characteristics of surface EMG signals. The results show that temporal jitter strongly reduces the extractability of spike train harmonics, while partial motor unit synchronization can restore structured spectral content. The Spectral Randomness Index (SRI) captures this transition from organized neural activity to stochastic EMG patterns, and the combined use of the FEI, SEI, MEI, and SRI provides a robust framework to assess neural and muscular contributions in the sEMG spectrum. By considering individual fiber positions, volume conduction, and tissue filtering, the analyses preserve key physiological and recording effects, highlighting how the interplay between neural timing and MUAP spectral envelopes governs observable EMG features.

These insights have potential implications for practical applications, including neurorehabilitation protocols and the assessment of corticomuscular coherence, where understanding how neural drive is expressed in the sEMG spectrum can inform intervention design [24,25]. Future work could leverage artificial intelligence and machine learning approaches to enhance the automated extraction of EMG features from the indices developed in this study, enabling predictive modeling, classification of motor unit behavior, or fatigue monitoring.

## Figures and Tables

**Figure 1 bioengineering-12-01181-f001:**
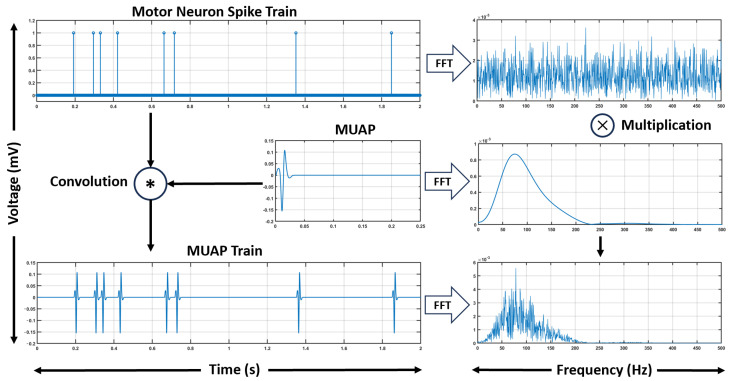
Illustration of the fundamentals of the modeling framework. The convolution of the spike train with the MUAP waveform produces the MUAP train in the time domain. The frequency spectra of the spike train and MUAP are obtained using the Fast Fourier Transform (FFT), and their convolution in the temporal domain corresponds to their multiplication in the frequency domain.

**Figure 2 bioengineering-12-01181-f002:**
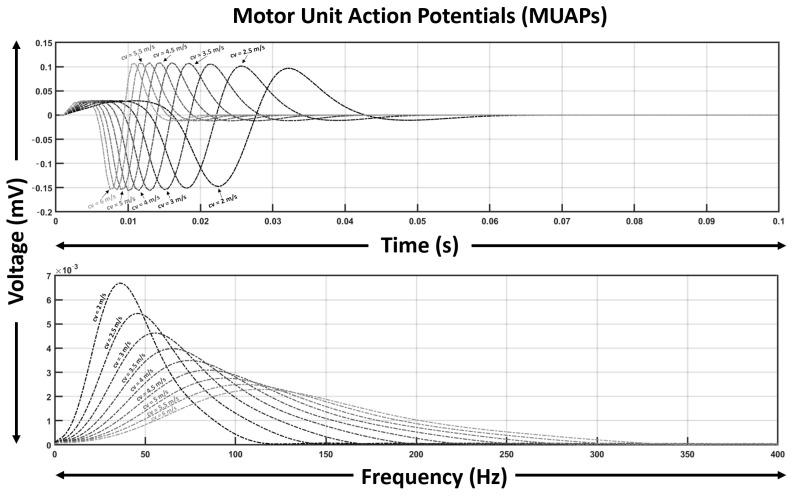
Simulated MUAP waveforms and their spectral distributions for mean conduction velocities ranging from 2 to 6 m/s. Mean conduction velocity of the fiber ensemble strongly modulates MUAP shape and spectral bandwidth.

**Figure 3 bioengineering-12-01181-f003:**
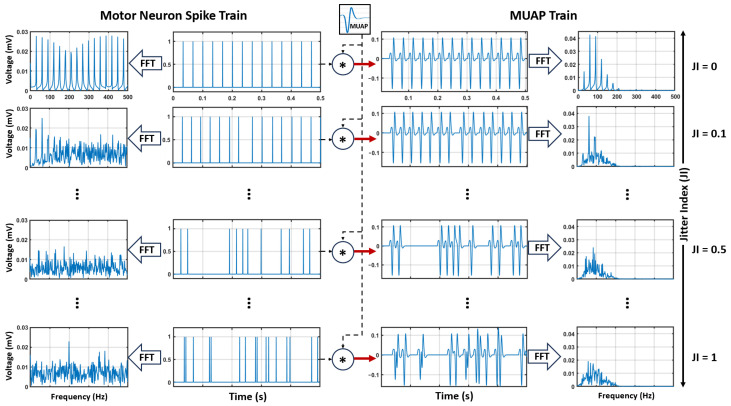
Effects of the Jitter Index (JI) on motor neuron spike trains and their spectral characteristics. Increased jitter results in greater spectral randomness and reduced prominence of average firing frequencies (**right side**). The resulting MUAP train spectrum illustrates the combined influence of spike train variability and MUAP shaping (**left side**).

**Figure 4 bioengineering-12-01181-f004:**
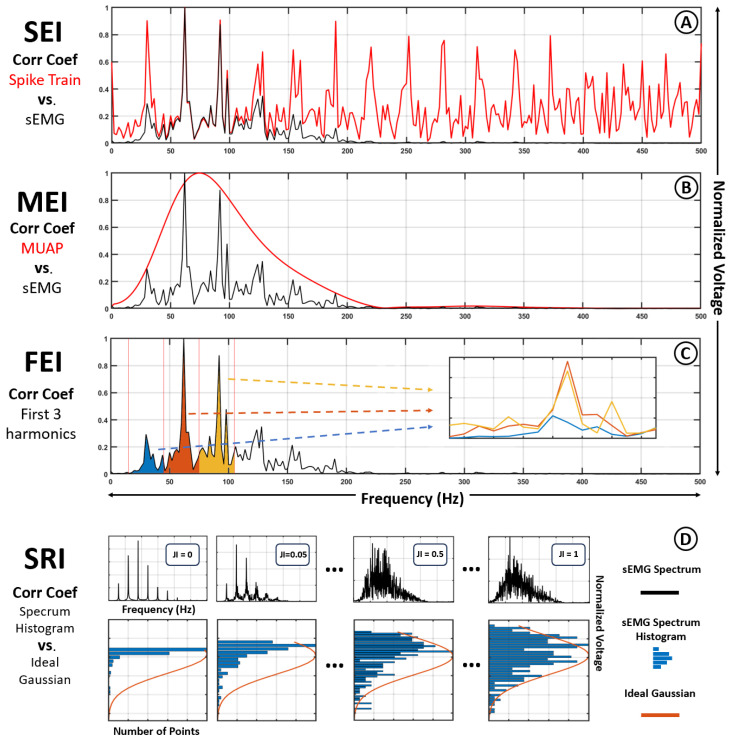
Visual representation of the comparisons performed to obtain each Extractability Index. (**A**) Spike train Extractability Index (SEI): Correlation coefficient between the sEMG spectrum (black) and the spike train spectrum (red). (**B**) MUAP Extractability Index (MEI): Correlation coefficient between the sEMG spectrum (black) and the MUAP spectrum (red). (**C**) Fire rate Extractability Index (FEI): Correlation coefficient between the first (blue), second (orange) and third (yellow) harmonics of the sEMG spectrum. (**D**) Spectral Randomness Index (SRI).

**Figure 5 bioengineering-12-01181-f005:**
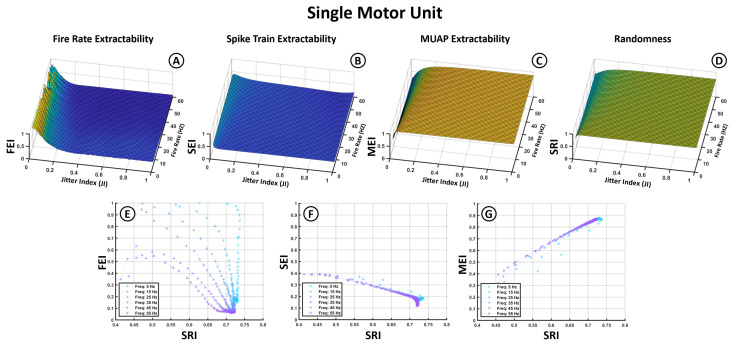
Top graphs represent the impact of jitter (X-axis) and firing frequencies (Y-axis) on the different indices (Z-axis) for individual motor units. (**A**) Fire Rate Extractability Index (FEI). (**B**) Spike Train Extractability Index (SEI). (**C**) MUAP Extractability Index (MEI). (**D**) Spectral Randomness Index (SRI). Bottom graphs represent the relation between (**E**) FEI, (**F**) SEI and (**G**) MEI (Y-axis) and the signal randomness quantified through the SRI (X-axis).

**Figure 6 bioengineering-12-01181-f006:**
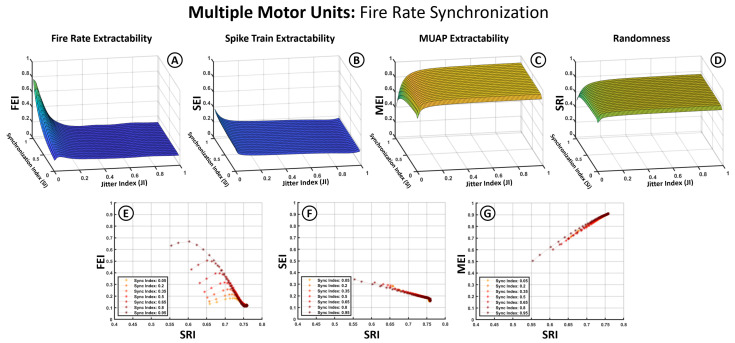
Effects of firing frequency-based synchronization on extractability indices for multiple motor units. (**A**–**D**) Three-dimensional surface plots illustrate the dependency of the SEI, FEI, MEI, and SRI (Z-axis) on the Jitter Index (X-axis) and the Synchronization Index (Y-axis). (**E**–**G**) graphs show the relation between the SEI, FEI, MEI (Y-axis) and the changes in signal randomness measured through the SRI (X-axis).

**Figure 7 bioengineering-12-01181-f007:**
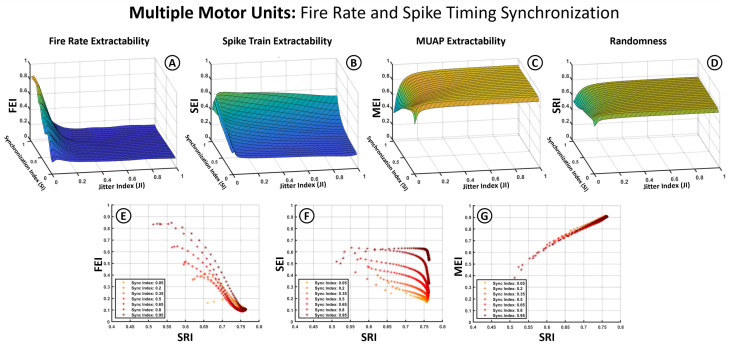
Influence of synchronization based on both firing frequencies and spike timing alignment on extractability indices for multiple motor units. (**A**–**D**) Three-dimensional surface plots illustrate the dependency of the SEI, FEI, MEI, and SRI (Z-axis) on the Jitter Index (X-axis) and the Synchronization Index (Y-axis). (**E**–**G**) graphs show the relation between the SEI, FEI, MEI (Y-axis) and the changes in signal randomness measured through SRI (X-axis).

## Data Availability

All the information needed for the replicability of results is contained within the paper. Appendix B includes the Matlab code necessary for the simulation of motor neuron spike trains with varying fire rates, jitter and synchronization levels.

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
