# Peer review of "The Spectral Footprint of Neural Activity: How MUAP Properties and Spike Train Variability Shape sEMG"

_bioengineering, 2025, doi:10.3390/bioengineering12111181_

Round 1

Reviewer 1 Report

Comments and Suggestions for Authors

This manuscript presents a high-quality, systematic modeling study of the interaction between neural control and the sEMG. The work successfully addresses a critical gap in understanding how intrinsic physiological variability (such as temporal jitter and synchronization) translates into the spectral characteristics observed at the skin surface.

The methodology is robust, and the findings provide much-needed quantitative support for the conservative view in the scientific community that spectral analysis of sEMG has significant limitations in accurately inferring neural control strategies.

The work is well-executed, logically structured, and the conclusions are clearly supported by the model results.

Author Response

Comment: This manuscript presents a high-quality, systematic modeling study of the interaction between neural control and the sEMG. The work successfully addresses a critical gap in understanding how intrinsic physiological variability (such as temporal jitter and synchronization) translates into the spectral characteristics observed at the skin surface.

The methodology is robust, and the findings provide much-needed quantitative support for the conservative view in the scientific community that spectral analysis of sEMG has significant limitations in accurately inferring neural control strategies.

The work is well-executed, logically structured, and the conclusions are clearly supported by the model results.

Response: We sincerely thank the reviewer for their positive and encouraging evaluation of our work.
We greatly appreciate the acknowledgment of the study’s methodological rigor and its contribution to clarifying the relationship between neural variability and the spectral features of sEMG signals.
We are encouraged by this feedback and will continue to build upon these findings in future research.

Reviewer 2 Report

Comments and Suggestions for Authors

1.This manuscript performed spectral analysis of MUAP and Spike trains by infusing jitter index.

2.The abstract should be revised to explicitly state the need of primary experimental aim. It remains unclear whether the experiment was designed to assess muscle fatigue, workload, recovery, decomposition or other physiological responses.

3. The manuscript primarily validates previously established methods rather than introducing novel techniques. The real findings indicate that the simulated and experimental sEMG analyses-through MUAP modeling, spike train simulations, and motor unit indices (FEI, SEI, MEI, SRI)-demonstrate predictable trends in motor unit behavior, jitter, and signal modulation. These results confirm the reliability of the modeling framework under different physiological conditions, such as firing rate variability and conduction velocity ranges.

4. The sEMG is syntheticaly generated using mathematical model of temporal convolution using two components such as spike trains and MUAP. There is no real time dataset is used for perfect analysis.

5. The impact of skin and tissue impedance can significantly alter the amplitude, shape, and timing of Motor Unit Action Potentials (MUAPs). The manuscript should discuss how these physiological and recording factors were controlled or compensated for during data acquisition. Including references addressing this effect would strengthen the methodological rigor.

6. The real sEMG recordings are often contaminated by motion artifacts, crosstalk from neighboring MUAPs, and power-line interference, all of which can significantly distort the signal. It is unclear whether the synthetic sEMG model accounts for these realistic noise conditions. The authors should clarify whether such artifacts were simulated or filtered. If so, describe the noise modeling approach adopted.

7. The study employs a monopolar electrode configuration for sEMG acquisition. However, the rationale for selecting this method over the bipolar configuration is not clearly justified. Bipolar recording is generally preferred in experimental and clinical sEMG studies due to its superior common-mode noise rejection, reduced motion artifacts, and improved signal-to-noise ratio. The authors should clarify why the monopolar method was chosen.

8. L-130.  MUAPs are simulated across an extended range of mean conduction velocities (2–6 m/s). This range encompasses typical values for different fiber types (typically 3–5 m/s) while also covering variability introduced by anatomical and recording conditions. However, no reference is provided to justify this selection.

9. L-100. The multiplication of both spectra causes the spike train spectrum to be modulated by the MUAP spectrum, which acts as an envelope. The envelop will assumes simple multiplication (linear modulation), while in reality, muscle tissue and electrode-skin interfaces introduce nonlinear effects. Address this in manuscript.

10. L-175: The manuscript states that the jitter index was calculated using a firing rate of 30 Hz. However, the rationale for selecting this particular frequency is not explained. The authors should clarify why 30 Hz was chosen—whether it reflects typical motor neuron firing rates for the studied muscle, aligns with previous literature, or is optimal for evaluating jitter sensitivity. Providing justification or referencing standard protocols would strengthen the methodological clarity.

11. Figure 5 presents the relationships between FEI, SEI, MEI, and SRI versus the Jitter Index, as well as FEI, SEI, and MEI versus SRI. However, the key conclusions or insights from this analysis are not clearly summarized. Author need to highlight how these indices interact and what they reveal about muscle activation, motor unit behavior, or signal quality. 

12. Figures 6 and 7 illustrate the trade-offs among multiple motor units. However, the manuscript does not clearly state the key findings or conclusions derived from these relationships.

13. The manuscript does not utilize artificial intelligence (AI) or machine learning (ML) approaches, despite their potential to enhance the analysis of sEMG signals and motor unit behavior. Incorporating AI/ML could enable more sophisticated pattern recognition, predictive modeling of fatigue, or automated classification of motor unit activity, which would significantly strengthen the study’s novelty and practical impact. 

14. However, the study could be strengthened by explicitly linking these findings to practical applications, guiding neurorehabilitation protocols. 10.1109/EMBC40787.2023.10340505 and 10.1109/EMBC53108.2024.10781868 

15. Provide conclusion and future work for this research.

Author Response

Comment 1-2:
1. This manuscript performed spectral analysis of MUAP and Spike trains by infusing jitter index.

2. The abstract should be revised to explicitly state the need of primary experimental aim. It remains unclear whether the experiment was designed to assess muscle fatigue, workload, recovery, decomposition or other physiological responses.

Response: We thank the reviewer for this comment. We have revised the abstract to clearly state the primary aim of the study. As clarified, our work is not designed to assess specific experimental conditions such as fatigue, workload, or recovery. Instead, the main goal is to provide a quantitative framework to understand how variability in neural timing and muscle properties shapes the sEMG spectrum. This revision makes the scope and motivation of the study explicit.The revised abstract now includes:

“Rather than addressing a particular experimental condition such as fatigue or workload, the main goal of this study is to provide a framework that clarifies how variability in neural timing and muscle properties affects the observed sEMG spectrum. We introduce extractability indices to measure how clearly neural activity appears in the spectrum.”

Comment 3
3. The manuscript primarily validates previously established methods rather than introducing novel techniques. The real findings indicate that the simulated and experimental sEMG analyses-through MUAP modeling, spike train simulations, and motor unit indices (FEI, SEI, MEI, SRI)-demonstrate predictable trends in motor unit behavior, jitter, and signal modulation. These results confirm the reliability of the modeling framework under different physiological conditions, such as firing rate variability and conduction velocity ranges.

Response: We appreciate the reviewer’s comment. While the simulation framework builds upon established methods for MUAP and spike train modeling, the extractability indices (FEI, SEI, MEI, SRI) introduced in this study are novel metrics, providing a quantitative tool to assess the spectral visibility of neural and muscular components in sEMG signals.

Comment 4:
4. The sEMG is syntheticaly generated using mathematical model of temporal convolution using two components such as spike trains and MUAP. There is no real time dataset is used for perfect analysis.

Response: We acknowledge that this study relies on synthetically generated sEMG signals rather than real-time experimental datasets. However, obtaining detailed information from the spike trains of multiple motor neurons in vivo remains technically challenging and prone to errors in component extraction. Additionally, in experimental recordings it is not possible to systematically control parameters such as motor unit synchronization or spike timing jitter. Our simulation framework allows us to precisely manipulate these factors, providing clear insights into how variations in firing patterns and MUAP properties affect the resulting sEMG spectrum. Indeed, our results suggest that high jitter or synchronization variability could further complicate decomposition in real data. By using physiologically grounded simulated signals, we offer a controlled yet effective approach to understand the interaction between neural and muscular spectral components.

We added the following clarification in the Modelling Framework section:

"No real experimental data were used, but this approach allows precise control over spike timing, jitter, and synchronization, which is challenging or impossible in vivo. By simulating physiologically plausible conditions, this framework provides clear insight into how spike trains and MUAP spectra interact, avoiding the measurement limitations and noise inherent in real sEMG recordings."

Comment 5:
5. The impact of skin and tissue impedance can significantly alter the amplitude, shape, and timing of Motor Unit Action Potentials (MUAPs). The manuscript should discuss how these physiological and recording factors were controlled or compensated for during data acquisition. Including references addressing this effect would strengthen the methodological rigor.

Response: We thank the reviewer for highlighting the importance of skin and tissue impedance on MUAP morphology. In our study, these effects are explicitly accounted for in the simulation framework. The model by Costa et al. (2025) incorporates the spatial distribution of muscle fibers as well as the filtering effects of overlying biological tissues, including skin, fat, and connective tissue. Each fiber action potential is propagated and attenuated according to its depth, orientation, and position relative to the electrode, allowing us to simulate realistic distortions in amplitude, timing, and waveform shape caused by volume conduction. By simulating MUAPs across a physiologically relevant range of conduction velocities and fiber configurations, our framework captures both variability and residual spectral effects arising from anatomical and recording conditions. This approach ensures that key physiological and recording factors are integrated into the model, providing a realistic and controlled basis for studying how neural spike trains and MUAPs interact in the frequency domain. We add a few lines in the Modeling MUAP section to emphasize this point:

"By explicitly modelling individual fiber positions and the volume conduction environment, including skin and tissue layers, while parameterizing MUAP morphology primarily via the ensemble mean conduction velocity, the model preserves key spatial and conductor effects (including nonlinear distortions at the skin surface) relevant for spectral analysis. Importantly, the effects of volume conduction are resolved using finite element modeling (FEM), which has been proposed as an effective methodology to account for electromagnetic field changes arising from the non-arbitrary distribution of biological tissues. Despite this complexity, the framework remains tractable for systematic investigation of how spike-train spectra and MUAP bandwidth interact in sEMG signals."

Comment 6:
6. The real sEMG recordings are often contaminated by motion artifacts, crosstalk from neighboring MUAPs, and power-line interference, all of which can significantly distort the signal. It is unclear whether the synthetic sEMG model accounts for these realistic noise conditions. The authors should clarify whether such artifacts were simulated or filtered. If so, describe the noise modeling approach adopted.

Response: We thank the reviewer for the comment regarding noise, crosstalk, and motion artifacts. In the present study, these sources of signal contamination were not explicitly modeled. The main focus of our work is to investigate the effects of spike-train temporal variability (jitter) and motor unit synchronization on sEMG spectral characteristics. Including additional noise sources or crosstalk would substantially increase the dimensionality of the analysis and make it difficult to isolate the contributions of jitter and synchronization.

We assume that the theoretical framework will be applied under experimental conditions designed to minimize these artifacts. Specifically, measurements should be performed with electrodes positioned according to standard anatomical guidelines (e.g., SENIAM) and under isometric contraction conditions, where motion artifacts are minimal. Appropriate filtering and signal preprocessing can further reduce residual noise. Our simulations thus provide a clean baseline for interpreting sEMG spectra, which can then inform experimental design and analysis in practical applications.

To clarify the scope and limitations of the model, we have added a Modelling Limitations subsection, which explicitly addresses these assumptions and the sources of contamination that are not simulated.

Comment 7:
7. The study employs a monopolar electrode configuration for sEMG acquisition. However, the rationale for selecting this method over the bipolar configuration is not clearly justified. Bipolar recording is generally preferred in experimental and clinical sEMG studies due to its superior common-mode noise rejection, reduced motion artifacts, and improved signal-to-noise ratio. The authors should clarify why the monopolar method was chosen.

Response: We thank the reviewer for this observation. The use of a monopolar electrode configuration in this study was motivated by its relevance to high-density surface EMG (HD-sEMG) systems, which are commonly employed for motor unit decomposition and spike-train extraction. In such systems, monopolar recordings are preferred because they preserve the spatial information necessary to separate individual motor unit action potentials across the electrode grid. Although bipolar configurations provide better common-mode noise rejection, they spatially filter the signal and reduce the ability to distinguish contributions from different motor units. To clarify this rationale, we have revised the manuscript to explicitly state that a monopolar configuration was selected to emulate the conditions of HD-sEMG decomposition studies (see revised text in the MUAP Modelling section).

Comment 8:
8. L-130.  MUAPs are simulated across an extended range of mean conduction velocities (2–6 m/s). This range encompasses typical values for different fiber types (typically 3–5 m/s) while also covering variability introduced by anatomical and recording conditions. However, no reference is provided to justify this selection.

Response: We thank the reviewer for the observation. A reference has been added to support the selected range of mean conduction velocities which reflects typical physiological variability across fiber types.

Comment 9:
9. L-100. The multiplication of both spectra causes the spike train spectrum to be modulated by the MUAP spectrum, which acts as an envelope. The envelop will assumes simple multiplication (linear modulation), while in reality, muscle tissue and electrode-skin interfaces introduce nonlinear effects. Address this in manuscript.

Response: We thank the reviewer for this insightful comment. While the convolution between spike trains and MUAPs is indeed a linear operation, the MUAPs in our model are not idealized or purely linear waveforms. They are simulated at the skin-surface level using the model by Costa et al. (2025), which explicitly incorporates the spatial distribution of fibers and the nonlinear filtering effects of the intervening tissues and skin. As a result, the nonlinearities associated with the volume-conductor properties and electrode–skin interface are implicitly embedded in the MUAP morphology and spectrum. Thus, the modulation observed in our results reflects both the linear convolution process and the underlying nonlinear distortions inherent to surface EMG signal generation. We have added a clarification in the manuscript to emphasize this point as pointed out in the reply to comment 5:

"By explicitly modelling individual fiber positions and the volume conduction environment, including skin and tissue layers, while parameterizing MUAP morphology primarily via the ensemble mean conduction velocity, the model preserves key spatial and conductor effects (including nonlinear distortions at the skin surface) relevant for spectral analysis. Importantly, the effects of volume conduction are resolved using finite element modeling (FEM), which has been proposed as an effective methodology to account for electromagnetic field changes arising from the non-arbitrary distribution of biological tissues. Despite this complexity, the framework remains tractable for systematic investigation of how spike-train spectra and MUAP bandwidth interact in sEMG signals."

Comment 10:
10. L-175: The manuscript states that the jitter index was calculated using a firing rate of 30 Hz. However, the rationale for selecting this particular frequency is not explained. The authors should clarify why 30 Hz was chosen—whether it reflects typical motor neuron firing rates for the studied muscle, aligns with previous literature, or is optimal for evaluating jitter sensitivity. Providing justification or referencing standard protocols would strengthen the methodological clarity.

Response:We thank the reviewer for pointing out the need to clarify the rationale for selecting a firing rate of 30 Hz. In our study, the Jitter Index (JI) is systematically evaluated across the full physiological range of firing frequencies (5–60 Hz) as detailed in the Simulation Protocols section. The example shown at 30 Hz in Figure 5 was chosen solely for illustrative purposes—to visualize the effect of increasing jitter on the spike-train and MUAP-train spectra (since 30 Hz approximately corresponds to the midrange of typical motor neuron firing rates during moderate isometric contractions). We clarified this point in Modeling Motor Neuron Spike Trains Section. The conclusions of the study do not depend on this specific frequency value, as the analysis framework generalizes across the full simulated range. 

"The 30 Hz example shown here is selected solely for visualization purposes, as it represents the midrange of those frequencies commonly observed during moderate isometric contractions."

Comment 11-12:
11. Figure 5 presents the relationships between FEI, SEI, MEI, and SRI versus the Jitter Index, as well as FEI, SEI, and MEI versus SRI. However, the key conclusions or insights from this analysis are not clearly summarized. Author need to highlight how these indices interact and what they reveal about muscle activation, motor unit behavior, or signal quality. 

12. Figures 6 and 7 illustrate the trade-offs among multiple motor units. However, the manuscript does not clearly state the key findings or conclusions derived from these relationships.

Response: We thank the reviewer for pointing out that the key insights from Figures 5–7 were not clearly summarized. In response, we have revised the Discussion section to explicitly synthesize the main findings and highlight the physiological and signal-processing implications of our analyses.

For Comment 11, we clarified how the Spectral Randomness Index (SRI) captures the transition from structured neural activity to stochastic EMG content, linking jitter-induced desynchronization to measurable reductions in spectral extractability. The revised text now reads:

"Together, these results show that the SRI effectively captures the transition from structured neural activity to stochastic EMG content, linking jitter-induced desynchronization to measurable losses in spectral extractability."

For Comment 12, we emphasized the interplay between multiple motor units, demonstrating the trade-off between jitter and partial synchronization, and how coordinated activity can restore structured spectral content. The revised text now reads:

"Hence, Figures \ref{fig6} and \ref{fig7} reveal a fundamental trade-off: while jitter introduces noise-like dispersion, partial synchronization can restore structured spectral content, illustrating how coordinated neural activity can enhance EMG interpretability."

These modifications synthesize the results across Figures 5–7, making explicit the relationships between the indices (FEI, SEI, MEI, SRI) and their implications for muscle activation, motor unit behavior, and signal quality.

Comments 13-15:
13. The manuscript does not utilize artificial intelligence (AI) or machine learning (ML) approaches, despite their potential to enhance the analysis of sEMG signals and motor unit behavior. Incorporating AI/ML could enable more sophisticated pattern recognition, predictive modeling of fatigue, or automated classification of motor unit activity, which would significantly strengthen the study’s novelty and practical impact. 

14. However, the study could be strengthened by explicitly linking these findings to practical applications, guiding neurorehabilitation protocols. 10.1109/EMBC40787.2023.10340505 and 10.1109/EMBC53108.2024.10781868 

15. Provide conclusion and future work for this research.

Response: We thank the reviewers for highlighting the potential to extend our work toward practical applications and AI/ML-based analyses. In response, we have added a new section, Conclusions and Future Work, which summarizes the key insights of the study and outlines future directions. Specifically, this section emphasizes how temporal jitter and partial motor unit synchronization affect spectral extractability, and how the Spectral Randomness Index (SRI) together with FEI, SEI, and MEI can be used to assess neural and muscular contributions in sEMG signals.

Moreover, we discuss the potential translational applications of our findings, including neurorehabilitation protocols and assessment of corticomuscular coherence. We also note that future work could leverage artificial intelligence and machine learning approaches to automate the extraction of EMG features from these indices, enabling predictive modeling, classification of motor unit behavior, or fatigue monitoring. This addition addresses the reviewers’ suggestions by linking the study’s outcomes to concrete applications and outlining AI/ML as a promising future direction.

Reviewer 3 Report

Comments and Suggestions for Authors

The paper provides a model that describes how the spectral characteristics of sEMG signals are derived from the convolution between neural impulse trains and MUAPs. To strengthen the document, the authors need to integrate and respond to the following observations.

1. Could the presented model be extended within a hybrid FEM-AI approach that integrates sEMG with thermographic or mechanical data to better describe neuromuscular energy transfer?

2. The paper demonstrates that spectral indices respond to jitter and synchronisation, but it doesn't clarify whether these behaviours can be observed in physiological recordings. How can the authors validate these simulated models with measured EMG datasets? Alternatively, the authors must integrate the Discussion with the study doi: 10.3390/electronics14112268, which demonstrates how FEM simulations of muscle tissue can be experimentally cross-validated with biosignal acquisitions during rehabilitation tasks. 

3. The study models synchronisation within a set of motor units, but real EMG recordings involve spatially distributed muscle groups and multichannel sensors. To this end, can the authors extend the synchronisation model to simulate intermuscular coordination or bilateral coupling?

4. The convolutional model takes into account volumetric conduction but assumes ideal electrode-skin coupling. How could the presence of non-uniform impedance, electrode movement, or tissue heterogeneity affect the MUAP morphology and derived spectral indices?

5. The introduced indices (FEI, SEI, SRI) are theoretically important, but their potential for quantitative neuromuscular assessment is not fully discussed. Could these parameters serve as biomarkers to monitor motor recovery or fatigue in clinical or rehabilitation settings?

6. The manuscript could benefit from a brief discussion comparing the proposed convolutional spectral model with the hybrid FEM-AI paradigms used for biomedical diagnostics.

7. The convolutional model assumes a simplified MUAP template controlled by conduction velocity, neglecting fibre-type heterogeneity, tissue anisotropy, and electrode placement variability.
How do these simplifications affect the interpretation of spectral results? Could a more extensive description of tissue conductivity based on finite elements provide a more realistic estimate of MUAP filtering effects?

8. The model assumes ideal volume conduction and perfect electrode coupling. The authors evaluated how experimental imperfections would modify the spectral indices, or whether these effects could be incorporated to better approximate real acquisition conditions.

9. The results show that the temporal synchronisation of spikes increases extractability indices, but the physiological mechanism remains unclear. Is the synchronisation the coupling of motor units observed in vivo, or is it an emergent computational artefact of the simulation?

Author Response

The paper provides a model that describes how the spectral characteristics of sEMG signals are derived from the convolution between neural impulse trains and MUAPs. To strengthen the document, the authors need to integrate and respond to the following observations.

Comment 1,2,6,7,8
1. Could the presented model be extended within a hybrid FEM-AI approach that integrates sEMG with thermographic or mechanical data to better describe neuromuscular energy transfer?

2. The paper demonstrates that spectral indices respond to jitter and synchronisation, but it doesn't clarify whether these behaviours can be observed in physiological recordings. How can the authors validate these simulated models with measured EMG datasets? Alternatively, the authors must integrate the Discussion with the study doi: 10.3390/electronics14112268, which demonstrates how FEM simulations of muscle tissue can be experimentally cross-validated with biosignal acquisitions during rehabilitation tasks. 

6. The manuscript could benefit from a brief discussion comparing the proposed convolutional spectral model with the hybrid FEM-AI paradigms used for biomedical diagnostics.

7. The convolutional model assumes a simplified MUAP template controlled by conduction velocity, neglecting fibre-type heterogeneity, tissue anisotropy, and electrode placement variability.
How do these simplifications affect the interpretation of spectral results? Could a more extensive description of tissue conductivity based on finite elements provide a more realistic estimate of MUAP filtering effects?

8. The model assumes ideal volume conduction and perfect electrode coupling. The authors evaluated how experimental imperfections would modify the spectral indices, or whether these effects could be incorporated to better approximate real acquisition conditions.

Response: We thank the reviewer for these valuable suggestions. While the present study focuses on the spectral interaction between spike trains and MUAPs and does not address thermodynamic or mechanical aspects of neuromuscular energy transfer, the modeling of MUAPs employed in Costa et al. \cite{costa2025tailoring} already leverages finite element modeling (FEM) to account for nonlinear filtering of electrical signals through the non-arbitrary distribution of biological tissues. We have explicitly added this information in the revised manuscript (MUAP Modelling Section), emphasizing that FEM provides a rigorous framework to resolve volume conduction effects, tissue anisotropy, and spatial heterogeneity, thereby improving the physiological realism of the simulated MUAPs and their derived spectral indices. This addition also strengthens the connection between the convolutional spectral model and FEM-based approaches in the literature \cite{lagana2025fem}, highlighting the potential for FEM to complement future experimental validation of spectral behaviours in measured sEMG signals.

Comment 3-4:
3. The study models synchronisation within a set of motor units, but real EMG recordings involve spatially distributed muscle groups and multichannel sensors. To this end, can the authors extend the synchronisation model to simulate intermuscular coordination or bilateral coupling?

4. The convolutional model takes into account volumetric conduction but assumes ideal electrode-skin coupling. How could the presence of non-uniform impedance, electrode movement, or tissue heterogeneity affect the MUAP morphology and derived spectral indices?

Response: We thank the reviewer for highlighting these important considerations. The current study focuses on controlled, isometric contraction conditions with electrodes positioned according to SENIAM guidelines, which minimizes motion artifacts, cross-talk, and other sources of recording noise. Consequently, the model does not explicitly simulate electrode-skin coupling variations, non-uniform impedance, electrode movement, tissue heterogeneity, or intermuscular/bilateral coordination. These simplifications represent current limitations of the model, but they were intentionally chosen to isolate the effects of spike timing, jitter, and intra-unit synchronization. We acknowledge that in more complex or dynamic experimental scenarios, these factors could affect MUAP morphology and spectral indices. To address this, the Modeling Limitations subsection has been added to the manuscript to clarify these points.

Comment 5:
5. The introduced indices (FEI, SEI, SRI) are theoretically important, but their potential for quantitative neuromuscular assessment is not fully discussed. Could these parameters serve as biomarkers to monitor motor recovery or fatigue in clinical or rehabilitation settings?

Reponse: We thank the reviewer for highlighting the potential translational relevance of the proposed indices (FEI, SEI, SRI). In response, we have added statements in the Discussion to explicitly summarize the main implications of these indices, emphasizing how they capture the transition from structured neural activity to stochastic EMG content and their relationships with extractability metrics. Furthermore, in the newly added Conclusions and Future Work section, we highlight potential applications of these indices.

Comment 9:
9. The results show that the temporal synchronisation of spikes increases extractability indices, but the physiological mechanism remains unclear. Is the synchronisation the coupling of motor units observed in vivo, or is it an emergent computational artefact of the simulation?

Response:
We thank the reviewer for this insightful observation. The synchronization modeled in this study represents the common synaptic input to motor neurons observed experimentally in vivo, rather than an emergent computational artifact. Several electrophysiological studies have demonstrated that groups of motor neurons can receive correlated synaptic drive, resulting in partial synchronization of their discharge times. In our simulations, this physiological phenomenon was parameterized by introducing a controlled degree of temporal coupling among motor unit spike trains, allowing systematic investigation of how synchronization strength modulates spectral indices in the sEMG. No artificial correlation structures were imposed beyond this physiologically motivated mechanism.
We have additionally cited relevant experimental studies reporting synchronization of motor neuron activity in real sEMG recordings to clarify the physiological basis of the modeled behavior.

Round 2

Reviewer 3 Report

Comments and Suggestions for Authors

I thank the authors for their replies to the comments. I have no further comments to make.